# Effect of Deposition Temperature on the Structure, Mechanical, Electrochemical Evaluation, Degradation Rate and Peptides Adhesion of Mg and Si-Doped Hydroxyapatite Deposited on AZ31B Alloy

Anca Constantina Parau [1], Mihaela Dinu [1], Cosmin Mihai Cotrut [2], Iulian Pana [1], Diana Maria Vranceanu [2], Lidia Ruxandra Constantin [1], Giuseppe Serratore [3], Ioana Maria Marinescu [1], Catalin Vitelaru [1], Giuseppina Ambrogio [3], Dennis Alexander Böhner [4], Annette G. Beck-Sickinger [4] and Alina Vladescu (Dragomir) [1,5,*]

1 Department for Advanced Surface Processing and Analysis by Vacuum Technologies, National Institute of Research and Development for Optoelectronics—INOE 2000, 77125 Magurele, Romania; lidia.constantin@inoe.ro (L.R.C.)

2 Department of Metallic Materials Science, Physical Metallurgy, Faculty of Materials Science and Engineering, University Politehnica of Bucharest, 60042 Bucharest, Romania

3 Department of Mechanical Energy and Management Engineering, University of Calabria, 87036 Rende, Italy

4 Institute of Biochemistry, Faculty of Life Sciences, Leipzig University, D04316 Leipzig, Germany; dennis.boehner@uni-leipzig.de (D.A.B.)

5 Physical Materials Science and Composite Materials Centre, Research School of Chemistry & Applied Biomedical Sciences, National Research Tomsk Polytechnic University, 634050 Tomsk, Russia

* Correspondence: alinava@inoe.ro; Tel.: +40-21-457-57-59

**Abstract:** Degradable and non-degradable biomaterials are two categories that can be used to classify the existing biomaterials, being a solution for eliminating a second surgical intervention of the implant when the tissue has properly recovered. In the present paper, the effect of deposition temperature on the structure, morphology, hardness, electrochemical evaluation, degradation properties and functional peptides adhesion of Mg and Si-doped hydroxyapatite was investigated. The coatings were obtained by RF magnetron sputtering technique at room temperature (RT) and 200 °C on AZ31B alloy substrate. Results showed that an increase in deposition temperature led to an improvement in hardness and reduced modulus of about 47%. From an electrochemical point of view, a comparative assessment of corrosion resistance was made as a function of the immersion medium used, highlighting the superior behaviour revealed by the coating deposited at elevated temperature when immersed in DMEM medium ($i_{corr} \sim 12\ \mu A/cm^2$, $R_{coat} = 705\ \Omega\ cm^2$, $R_{ct} = 7624\ \Omega\ cm^2$). By increasing the deposition temperature up to 200 °C, the degradation rate of the coatings was slowed, more visible in the case of DMEM, which had a less aggressive effect after 14 days of immersion. Both deposition temperatures are equally suitable for further bio-inspired coating with a mussel-derived peptide, to facilitate biointegration.

**Keywords:** magnetron sputtering; hydroxyapatite; hardness; corrosion; degradation; peptides adhesion

## 1. Introduction

Degradable and non-degradable biomaterials are two categories that can be used to classify existing biomaterials. Researchers have shown a greater interest in biodegradable materials in recent years compared to more typical biologically inert metal materials, such as stainless steel, titanium alloy, cobalt-based alloy, and so on [1–6]. The necessity for a second surgical intervention needed to remove the implant when the tissue has properly recovered is eliminated when using biodegradable implants, which is one of the most significant advantages of these types of devices [7,8].

In the topic of magnesium and magnesium alloys, more than 4000 papers were published in 2021 alone, illustrating that this is a hot spot in the materials science and engineering field [9]. Also, there has been an increasing amount of focus in terms of developing magnesium alloys as a viable treatment solution for orthopaedic injuries and cardiovascular diseases. Nevertheless, the quick rate of magnesium breakdown continues to be a significant barrier to the widespread utilisation of magnesium. Therefore, the development of biodegradable magnesium alloys with tuneable degradation rates is crucial [4,10–12].

As a potential material for use in biomedicine, magnesium alloy possesses several benefits. It is one of the primary metal elements found in the human body, and its levels are second only to those of calcium, sodium, and potassium, being a crucial component for bone development, favouring cell proliferation [3,10,13]. Also, it is involved in 300 enzymatic reactions [5]. Alloys made of magnesium have mechanical characteristics that are comparable to those of human bones [4,10,14]. In the long-axis direction, the tensile strength, young's modulus, and density of human cortical bone are as follows: 120–150 MPa, 20 GPa, and 1.8 g/cm$^3$, respectively. In comparison, the mechanical properties of magnesium alloy are as follows: 200–300 MPa, 40–45 GPa, and 1.74 g/cm$^3$, respectively. In addition, the elastic modulus is somewhere in the range of 41–45 GPa, which is very close to that of the human bone [7,10,15]. Also, the low density of the engineering metals of magnesium is approximately 65% as dense as aluminium alloys, 38% as dense as titanium, and 25% as dense as steel [16,17]. As the structural metal with the lowest density, magnesium is widely regarded as the ideal candidate for use in modern alloys [18].

The fundamental principles of degradation mechanism and corrosion products have been largely acknowledged during the many years of study conducted worldwide on magnesium alloys. When it comes to magnesium alloys, the primary areas of focus are still the control of the degradation rate and the behaviour of ion release [12,18,19].

In 1878, vascular ligation with magnesium alloys was performed for the first time, and since then, the benefit of using biodegradable materials in medicine was continuously demonstrated [7,10,15,18,20]. Recent research has resulted in the development of new magnesium zinc composites with further characteristics since zinc is another vitally important trace element for the human body. It is second only to iron in terms of its content in the human body, which comes in at roughly 2 g. According to the studies carried out so far, the presence of zinc within magnesium alloys has a significant impact on their resistance to corrosion [3,8,10].

The most difficult aspect of this situation is figuring out how to customise the degrading process in a manner that is appropriate for a biological setting. Surface treatment is one of the primary tactics that has been extensively researched as a way of modifying the mechanical characteristics of magnesium and its alloys in order to slow down the degradation rate [5,19].

If researchers and engineers could forecast the rates of corrosion, they would be better able to develop materials with appropriate corrosion rates without significantly compromising the material's mechanical qualities [21]. Despite the continuous efforts undertaken to make the metallic materials more resistant to corrosion, which is an electrochemical process that includes both reduction and oxidation reactions, the underlying problem persists [19]. One suitable possibility to control the corrosion process and to prevent the body fluids from directly interacting with magnesium-based alloys is the usage of coatings, which act as protective layers at the body-magnesium implant interface [10,20].

Surface coatings can either drastically cut down the rate of localised degradation or, at the very least, postpone the rate at which magnesium-based materials are attacked locally. There are a few different coating processes available for magnesium and its alloys, through which ceramic and/or polymeric-based films can be obtained [6,12,22–26].

Among the material of choice as a coating for magnesium alloy substrate, one can find hydroxyapatite (HAp, $Ca_{10}(PO_4)_6(OH)_2$), which is a ceramic biomaterial, also allowing the addition of different doping elements to further favour the osteointegration and control the alloys corrosion rate [27,28]. In addition to this, it is known that calcium is one of the most

crucial nutrients, involved in several processes and mechanisms of the human body [13]. Because HAp possesses strong bioactivity and osteoconductivity, it can rapidly integrate with the bones and stimulate the formation of new hard tissue. This ability is critical for bone regeneration, as was reported in Refs. [22,29].

Coatings made of calcium phosphate-based materials, such as HAp, are non-toxic, osteoconductive, and have high biocompatibility [30]. As a result, a significant number of studies have concentrated on calcium phosphate coatings for use in biomedical applications such as bone substitutes and orthopaedics. A layer of hydroxyapatite, which is the primary component of natural bone, can occur through the presence of calcium and phosphorus elements [2,15,20]. There are a few review articles that discuss several coating solutions, including HAp coatings, that can be used on magnesium-based materials [16,22,30–32].

Surface modification of biomaterials can be accomplished by several different procedures, one of which is called magnetron sputtering. Using this method, hydroxyapatite coating properties like topography, Ca/P ratio, density, thickness, etc., can be modified by altering the sputtering parameters such as air pressure, applied voltage, substrate-to-target distance, and deposition time [1,14,16,33]. Magnetron sputtering results in coatings that are dense, have a strong adherence to metallic surfaces and have an elemental composition that can be controlled and tuned [34,35].

The interest in the addition of biocompatible elements has increased in medical applications, especially in the coatings field. For this study, silicon (Si) was used as a doping element in order to increase HAp's corrosion resistance since it acts as a barrier between the substrate and the body fluids. Silicon is a biocompatible element that sustains the function of osteogenic cells [14,22] and is essential for the formation and growth of bone, teeth, and other skeletal elements. Studies suggest that the addition of Si to HAp facilitates the precipitation of an apatite layer on the materials' surface in an artificial physiological solution [36,37]. Used as a doping element, magnesium helps the osteointegration process [37]. According to the findings of in-vitro degradation investigations, the presence of the coating has significantly improved the corrosion resistance of magnesium alloy and has increased the bioactivity [38].

The objective of this study was to investigate the effect of deposition temperature (room temperature vs. 200 °C) on the structure, morphology, hardness, electrochemical evaluation, degradation properties and peptides adhesion of Mg and Si-doped hydroxyapatite coatings obtained by RF magnetron sputtering to improve AZ31B alloy properties. The deposition temperature was selected based on the previous results published in Ref. [34], where it was reported that the best deposition temperature was 200 °C. According to the previously published results [34], the deposition temperature has a significant effect on the crystallinity, mechanical and degradation rate of HAp coatings. Based on this paper, it was found that the coatings deposited at RT have an amorphous structure, while those deposited at 200 °C were crystalline with good corrosion resistance in simulated body fluid (SBF). Both coatings proved to have proper mechanical properties compared to those deposited at 100, 300 or 400 °C. Thus, based on the reported results, the present research is focused on the coatings prepared at RT and 200 °C.

## 2. Experimental Details

### 2.1. Substrate

AZ31B was used as substrates for the present paper, being supplied in flat sheets with a nominal thickness of 1 mm made by cold rolling and final annealing treatment. The chemical composition is reported in Table 1.

**Table 1.** AZ31B chemical composition (wt.%) of the as-received alloy.

| Al | Zn | Mn | Fe | Cu | Si | Ni | Ca | Mg |
|----|----|----|----|----|----|----|----|----|
| 2.98 | 1.03 | 0.34 | 0.0022 | 0.0067 | 0.0089 | 0.00047 | - | Rest |

Prior to coating deposition, the specimens were processed via Single Point Incremental forming (SPIF) [39] to obtain samples with mechanical and superficial conditions more equivalent to those that could be obtainable in a real prosthetic usecase. A standard geometry, i.e., a pyramid with a 154 square base characterised by an inclined sidewall of 40°, a major base equal to 149 mm 155 and a final depth of 30 mm, was manufactured, and a set of small samples (10 × 10 mm$^2$) were extracted by each side of the pyramid for the subsequent coating. The SPIF process was achieved by a hemispherical head punch tool with a diameter of 10 mm and a step depth of 0.1 mm on a Mazak Nexus 410 milling machine equipped with a heating chamber able to keep constant the sheet temperature up to about 250 °C during the forming process [40]. A feed rate of 0.5 m/min and a spindle speed of 4000 rpm were used. A D321R Molykote spray was employed as a lubricant. The substrates were obtained by the side faces of the above-described geometry.

## 2.2. Coatings

RF magnetron sputtering deposition unit was used for preparing the coatings (AJA International, Scituate, MA, USA). The system was equipped with three cathodes (Φ 20.4 mm) made of HAp (99.99% purity), MgO (99.99% purity) and SiC (99.99% purity), positioned in a confocal geometry. To achieve a similar deposition rate as of the HAp, the addition of Mg and Si into HAp-based coatings was carried out with cathodes made of oxides. In the present study, the coatings were prepared on two types of substrates: silicon wafers with <111> orientation and AZ31B alloy, depending on the type of investigation technique. To ensure the uniformity of the coatings, the samples were positioned on a rotating holder (15 rpm) at a distance of 120 mm from the targets. Prior to each deposition run, the substrates were ultrasonically cleaned with isopropyl alcohol for 15 min in an ultrasonic bath, and consequently, they were sputter cleaned in Ar$^+$ plasma for 15 min (f = 13.56 MHz, $p$ = 50 W, U$_{bias}$ = −310 V, $p$ = 0.67 Pa, without any intentional heating on substrates holder) and the vacuum chamber was evacuated down to $1.4 \times 10^{-4}$ Pa. The deposition conditions are presented in Table 2.

**Table 2.** The deposition conditions: $p$—pressure, V$_{RF}$—RF bias on substrates, P—power applied on the cathode, T—deposition temperature.

| Coatings | $p$ (Pa) | V$_{RF}$ (V) | P$_{MgO}$ (W) | P$_{SiC}$ (W) | T (°C) |
|---|---|---|---|---|---|
| HAp + Mg + Si_RT | 0.67 | −60 | 25 | 15 | RT |
| HAp + Mg + Si_200 | 0.67 | −60 | 25 | 15 | 200 |

## 2.3. Investigation Techniques

An energy-dispersive X-ray spectrometer (EDS, Bruker, Billerica, MA, USA) was used to determine the elemental composition of coatings.

The coatings topography was evaluated by atomic force microscopy (AFM, Veeco-Bruker, Billerica, MA, USA), using tapping mode 1 × 1 μm$^2$ with a scan speed of 0.3 Hz.

Grazing Incidence X-ray diffraction (GIXRD) was carried out to evaluate the phase composition of the coating using a SmartLab diffractometer (Rigaku, Tokyo, Japan) at an incident angle of the X-ray beam to 3°.

The nanoindentation test was used for determining the hardness and elastic modulus of the coatings using a Hysitron Premier TI nanomechanical characterisation system (Bruker, Billerica, MA, USA). The nanoindenter was equipped with a 142.3° blunt Berkovich tip with a 100 nm curvature radius. The surface of the sample was also scanned with the same indenter tip before testing to ensure a smooth surface for nanoindentation. Before any indentation testing, potential sources of uncertainty and errors such as thermal drift, initial penetration depth, and machine compliance were considered. A standard fused quartz sample (H = 9.25 GPa ± 10%, E = 69.6 GPa ± 10%) was used to calibrate the load force. The contact depth of every single indentation point was larger than 40 nm to overcome the limitation given by the geometrical characteristics of the used Berkovich tip. A minimum

physical distance of 5 μm was established between at least 25 indentation points, while the time intervals for load, hold and unload were 7 s, 2 s and 7 s, respectively.

In vitro electrochemical tests in SBF (simulated body fluid) and DMEM (Dulbecco's Modified Eagle Medium) solutions were used for evaluating the corrosion resistance of the coatings using a PARSTAT 4000 Potentiostat/Galvanostat (Princeton Applied Research—Ametek, Oak Ridge, TN, USA). A conventional three-electrode electrochemical cell was employed: working electrode (WE, coated samples), calomel reference electrode (SCE) and Pt foil as a counter electrode (CE). The tests were performed in SBF (pH = 7.4, [41]) at $37 \pm 0.5$ °C, being held constant with a heated circulating bath (Jeio Tech, CW-05G, Yuseong-gu, Daejeon, Republic of Korea). The electrochemical tests were performed in good agreement with a protocol described in ASTM standard G5-94 (2014) as follows. The open circuit potential ($E_{OC}$) was monitored for 1 h, and the Tafel plots were carried out between $\pm 0.200$ mV vs. $E_{OC}$ at a scanning rate of 1 mV/s. The main electrochemical parameters—corrosion potential ($E_{corr}$) and corrosion current density ($i_{corr}$), were estimated by Tafel plots extrapolation, according to ASTM G59-97 standard (reapproved 2020) [42].

In vitro, electrochemical impedance spectroscopy (EIS, Princeton Applied Research—Ametek, Oak Ridge, TN, USA) measurements were performed over the frequency range of $0.1 \div 10^5$ Hz by applying a sinusoidal signal with an amplitude of 10 mV RMS vs. $E_{OC}$. The data were recorded by VersaStudio software (version 2.62.2, Princeton Applied Research, Oak Ridge, TN, USA), and the EIS fitting procedure was performed using ZView software (version 12136-4, Scribner Associates Inc., Southern Pines, NC, USA).

In vitro degradation tests were performed in SBF and DMEM for 1, 3, 7, and 14 days at $37 \pm 0.5$ °C, using an incubator (Memmert IF 55, Memmert GmbH, Büchenbach, Germany). The testing solutions were changed everytwo days to ensure the necessary ionic content and to prevent the growth/formation of bacteria or other microorganisms. After each period of immersion, each sample was washed with distilled water and then dried for 24 h in a desiccator to remove any remaining water from its surface. Then, the samples were weighed with an analytical balance to evaluate the sample's mass evolution [41,43]. Five samples of each coating were used for immersion assays, and the mass measurements were conducted five times for each sample. The mass evolution was calculated based on the formula: $\Delta m = mf - mi$ (mg), where: $\Delta m$ represents the mass variation; $mf$ is the sample's mass after immersion; $mi$ is the sample's initial mass.

A scanning electron microscope (SEM, TableTop 3030PLUS, Tokyo, Japan) was used to evaluate the coatings aspect after the electrochemical and biodegradation tests.

Decorating biomaterial surfaces with functional peptides is a promising approach to improve implant-host interactions. Immobilisation of the cell-adhesive motifs c [RGDfK] or the heparin-binding sequence FHRRIKA was successfully achieved by short bio-inspired peptides derived from mussel foot proteins [44,45]. The effect of the HAp-based coating on the surface binding ability of a similar mussel-derived peptide (MP) was investigated. To protect the uncoated backside from solvent exposure, samples were glued into a 12-well plate using silicone. For surface coating, samples were incubated with 1 μM peptide solution in TBS buffer (25 mM Tris, 137 mM NaCl, 2.7 mM KCl, pH 7.6) for 238 16 h shaking at room temperature. For SEM imaging, samples were incubated with unmodified MP, whereas a biotinylated peptide was used for the investigation of the surface binding affinity. The next day, the coating solution was aspired, and samples were washed twice with TBS-T buffer (25 mM Tris, 137 mM NaCl, 2.7 mM KCl, 0.1% Tween20, pH 7.6) as well as TBS buffer. After transferring and glueing samples into new wells, surfaces were first blocked with 10% BSA in TBS buffer for 20 min and then incubated with horseradish peroxidase-conjugated streptavidin (1:2000 in TBS containing 1% BSA) for 1 h. Subsequently, samples were washed four times with TBS-T buffer and detection of bound HRP-streptavidin conjugate was carried out using 3,3′,5,5′-tetramethylbenzidine (TMB). After 3 min the reaction was stopped with 1 M HCl and the absorption at 450 nm of the solution was measured (Infinite M200, Tecan). The result is shown as mean + standard error of the mean and represents two independent experiments performed in duplicates. Data were analysed using GraphPad

Prism 5.03 (GraphPad Software). Statistical significances were determined by one-way ANOVA following Dunnett's multiple comparison test.

## 3. Results

### 3.1. Elemental and Phase Compositions and Morphology

The EDS investigations were performed on the coatings deposited on the Si wafers substrate for determining the Mg content and on the Mg alloy substrate for determining the Si content. This research design was made in order to eliminate the influence of the Mg or Si element from each substrate. The chemical composition is presented in Figure 1a. The EDS spectrum of each coating is presented in Figure 1c. One may observe that Si and Mg content is up to ~7 at.%, indicating the formation of HAp-doped coatings. The Ca/P ratio was calculated at about 1.53 for the coating deposited at RT and 1.68 for those prepared at 200 °C. This finding showed that an increment of the deposition temperature assured the formation of a stoichiometric HAp structure. Usually, for the doped HAp coatings, the (Ca + Si + Mg)/P ratio is calculated, but in the case of the present paper, it was impossible to be evaluated because the substrates were Si or Mg alloy, having at least one element found in the coating. However, based on the EDS spectrum (Figure 1b), there are seen a small amount of some elements of bare Mg alloy, such as Zn. Other elements, such as Mn, Fe, Cu, and Ni, are too small to be detected by the EDS system.

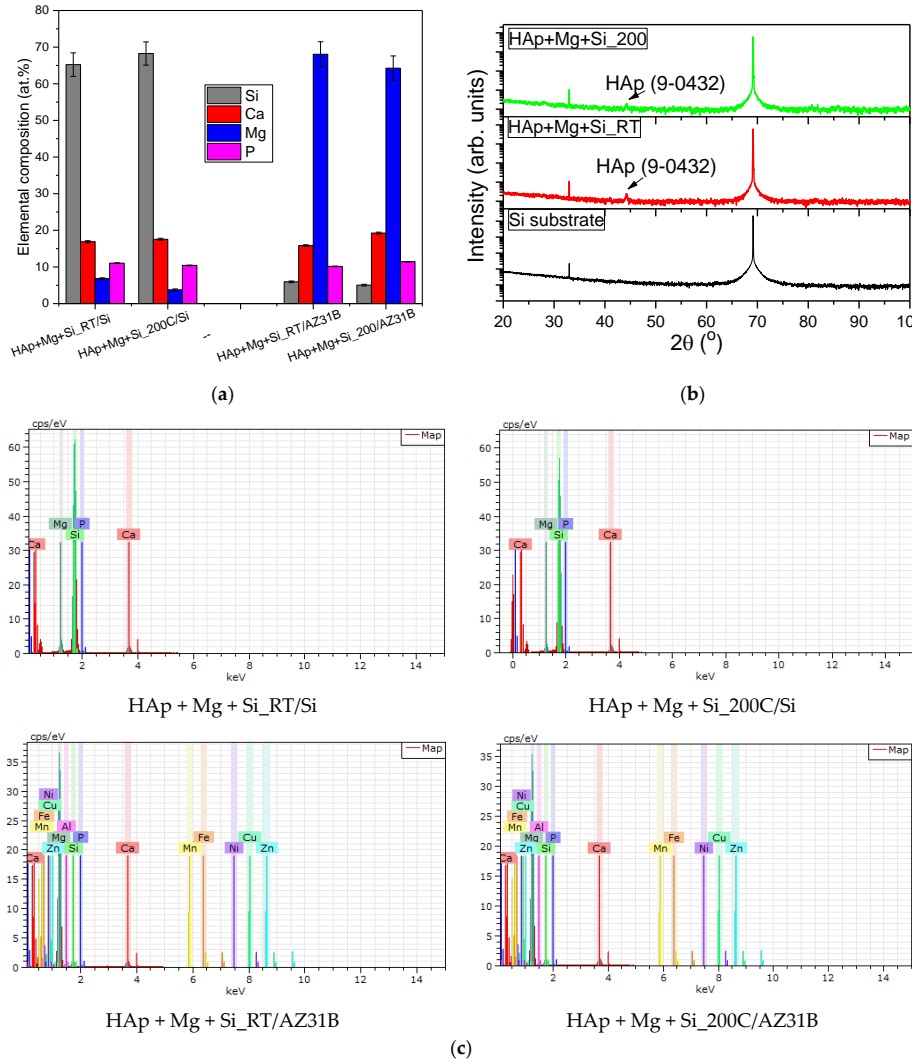

**Figure 1.** (**a**) Elemental composition; (**b**) GIXRD diffraction patterns of the coatings; and (**c**) EDS spectrum of each coating.

The GIXRD diffraction patterns of the coatings deposited on Si wafers are presented in Figure 1b. In both coatings, a small peak located at 44.32° can be found, being attributed to the HAp phase (JCPDS card no. 09-0432). This result indicates that the Si addition does not affect the formation of the HAp phase.

The morphology was evaluated based on AFM images (Figure 2). One can note that the morphology is quite similar for both coatings. Those prepared at 200 °C exhibited a denser structure with fine grains. It should be mentioned that the AFM images were recorded on coatings deposited on Si wafers to avoid disturbances from the roughness of the Mg alloy.

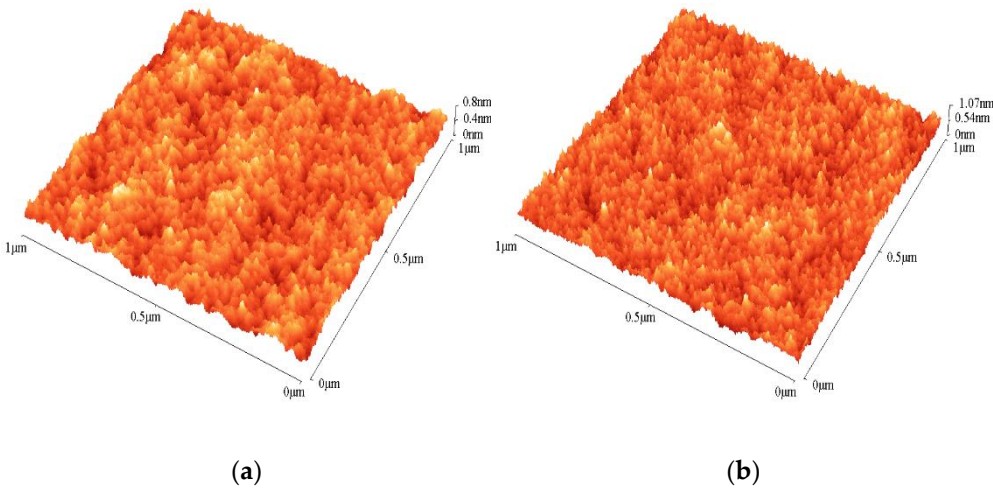

(**a**)          (**b**)

**Figure 2.** AFM images: (**a**) HAp + Mg + Si_RT and (**b**) HAp + Mg + Si_200 coatings deposited on Si waffers.

The 2D SEM images acquired for the coatings deposited directly on AZ31B alloys are presented in Figure 3. In both cases, the grooves' presence at 100× magnification is a common feature expected after the polishing process of AZ31B alloys before coating.

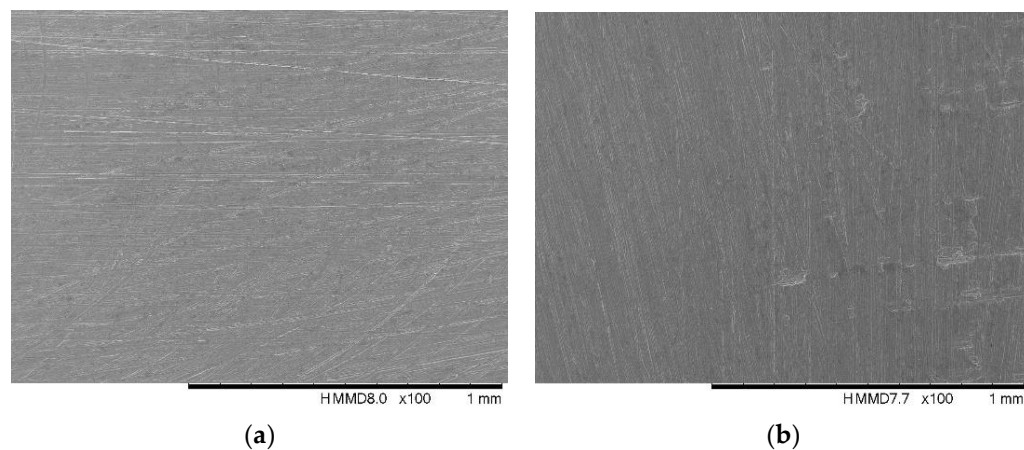

(**a**)          (**b**)

**Figure 3.** 2D SEM images: (**a**) HAp + Mg + Si_RT and (**b**) HAp + Mg + Si_200 coatings deposited on AZ31B substrates.

### 3.2. Nanoindentation

The mechanical properties of the coatings can significantly differ from those of bulk materials due to the confinement of the material at the nanometric scale. The unique properties of these thin layers are highly dependent on several factors such as the deposition method, thickness, structure, substrate type, or interface between substrate and layer. Therefore, it is important to adapt the deposition methods to obtain coatings that meet the specific technological requirements for smart functional and multifunctional biomaterials.

The hardness and reduced elastic modulus of the coatings were evaluated at penetration depths comparable to the coating's thicknesses. The results are presented in Figure 4. The substrate effect on the mechanical properties of the investigated layers was considered by employing a model that estimates the dependencies of hardness and reduced modulus as a function of substrate characteristics, maximum indentation depth and thickness [46]. The used model was only valid between the minimum 40 nm contact depth limit of the used nanomechanical system and a maximum penetration depth of ~90% of the film thickness. The maximum applied force used for nanoindentations was selected based on the coating's thicknesses and, in this case, had not exceeded a value of 2.5 mN.

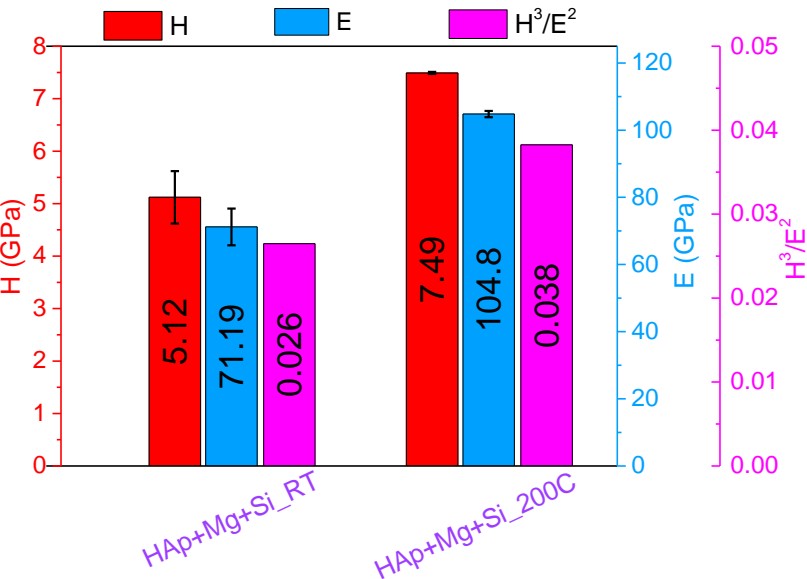

**Figure 4.** Derived hardness and reduced Young's modulus values of samples based on the Olive–Pharr algorithm.

The results indicated that the mechanical properties of the coatings differ significantly depending on the deposition temperature increment from RT to 200 °C. It was shown that both the hardness and reduced modulus of the coatings increase by approximately 47% as the deposition temperature increases. These results are consistent with our previous findings [47]. The selection of appropriate biomaterials as protective coatings for bare metallic implants is crucial and requires a balance between their hardness and friction properties [48]. One common parameter used to evaluate a material's resistance to plastic deformation is the $H^3/E_r^2$ ratio, which has also increased from 0.026 to 0.038. This suggests that the HAp + Mg + Si_200 °C sample may have superior resistance to plastic deformation when compared to the HAp + Mg + Si_RT sample.

### 3.3. In Vitro Corrosion Resistance

The Tafel plots of the investigated coatings obtained at room temperature and at 200 °C in SBF and DMEM testing media are given in Figure 5. The main electrochemical parameters obtained from the electrochemical tests are presented in Table 3. The values of the $i_{corr}$ parameter evolution as a function of the testing media for the investigated specimens are also shown in Figure 5.

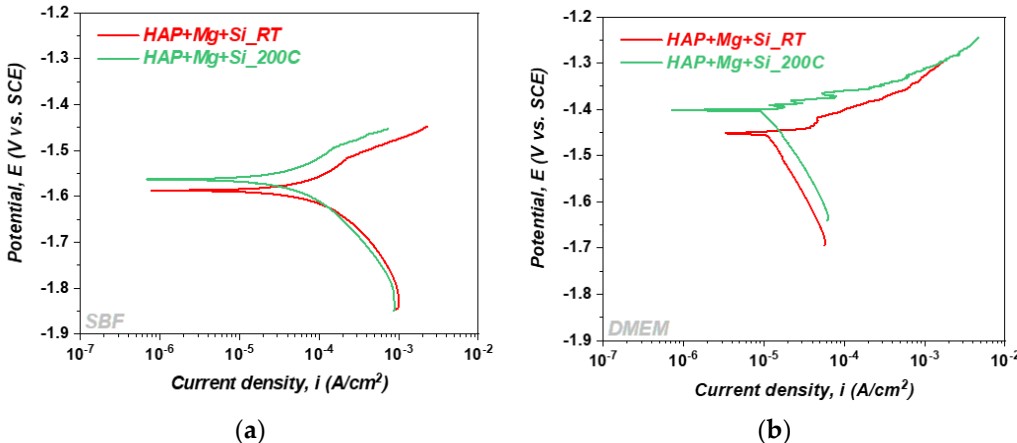

**Figure 5.** In vitro Tafel results: (**a**) Tests in SBF solution; (**b**) Tests in DMEM solutions.

**Table 3.** Electrochemical parameters of the coatings obtained in SBF and DMEM solutions.

| Coatings | Solutions | $E_{oc}$ (mV) | $E_{corr}$ (mV) | $i_{corr}$ ($\mu A/cm^2$) |
|---|---|---|---|---|
| HAp + Mg + Si_RT | SBF | −1669 | −1587 | 190.73 |
| | DMEM | −1510 | −1452 | 18.05 |
| HAp + Mg + Si_200 | SBF | −1668 | −1562 | 77.24 |
| | DMEM | −1511 | −1400 | 12.17 |

$E_{OC}$—open circuit potential, $E_{corr}$—corrosion potential, $i_{corr}$—current density.

It is well known that a material is more resistant to the hostility of a corrosive medium when the value of the corrosion potential ($E_{corr}$) has a more electropositive value and the corrosion current density ($i_{corr}$) has a smaller value. By considering this statement, it can be noticed that better values were reached for the coatings tested in the DMEM solution (Table 3 and Figure 5), indicating that both coatings are more resistant in DMEM. However, in both testing media and irrespective of the deposition temperature used, RT or 200 °C, the electrochemical parameters present a similar evolution.

In SBF, both coatings exhibited almost similar $E_{corr}$ values, while the $i_{corr}$ value decreased considerably (2.5 times) when the deposition temperature increased from RT to 200 °C. This finding shows that the coatings deposited at higher temperatures exhibited higher corrosion resistance in SBF.

In DMEM, the values of the $E_{corr}$ parameter for both coatings are close to each other, while the $i_{corr}$ value was found to be smaller for the coatings prepared at 200 °C, suggesting that an increment of the deposition temperature leads to higher corrosion resistance.

In a previous paper, we reported that the undoped HAp exhibited a value of the $i_{corr}$ from 23.07 $\mu A/cm^2$ (RT) to 12.23 $\mu A/cm^2$ (200 °C) in DMEM and from 439.18 $\mu A/cm^2$ (RT) to 129.16 $\mu A/cm^2$ (200 °C) in SBF, indicating that a temperature of 200 °C enhances the electrochemical behaviour of the undoped HAp based coatings. By comparing the parameters obtained for the present coating with those of the undoped HAp, it can be concluded that the addition of Si and Mg into the HAp matrix significantly improves the corrosion resistance in the SBF and DMEM solution of the AZ31B alloy.

### 3.4. In Vitro EIS Results

Figures 6 and 7 display the Nyquist and Bode plots of the investigated samples immersed in SBF and DMEM electrolytes after the stabilisation of the open circuit potential for 3600 s. Two defined time constants (indicated by arrows in Figures 6 and 7) characterised the HAp + Mg + Si samples, more visible in the case of the SBF test medium. Additionally, for both conditions, the Nyquist plot reveals the presence of an inductive loop in the low-frequency range (emphasised by a red square). According to Flores et al. [49], the time constant present in the medium frequencies range (MF) provides information related to coating–electrolyte interface, while the impedance data present in the low-frequency

range (LF) shows the occurrence of a corrosion process at the substrate–solution interface. Thus, the presence of an inductance loop in LF is often associated with the absorption of chloride ions through cracks and defects, causing the initiation of corrosion phenomena on the surface and, consequently, a decrease in substrate resistance [50]. Also, the high rate of Mg ions dissolution caused by low applied frequencies can be a responsible process as well [51].

Even though similar results were shown by the samples immersed in SBF [50,52,53], a slightly different behaviour was promoted by the electrolyte change. Higher semicircle diameters were obtained when DMEM was used, also confirmed by higher impedance modulus revealed by the Bode amplitude plot (Figure 7b). In this case, a more obvious difference was observed among the samples obtained at different deposition conditions, indicating an enhancement of dielectric properties influenced by the temperature increase. According to the literature, larger semicircle diameters are related to higher resistance to charge transfer, leading to a low corrosion rate of specimens under investigation [54].

Based on the impedance data displayed in Nyquist and Bode graphs, the presented electrical equivalent circuit was used to model the physical reactions (inset in Figures 6a and 7b). Thus, processes that occur within the analysed systems, such as fast (i.e., formation of the double layer, ohmic/charge transfer resistance) and slow processes (i.e., adsorbed species or transport phenomena), can be translated into quantitative information [55]. In this case, in addition to the often-used two-time constant equivalent circuit (with a CPE instead of ideal dielectric properties caused by inhomogeneity or current leakage [56]), L was also added, ascribed to corrosion product formation.

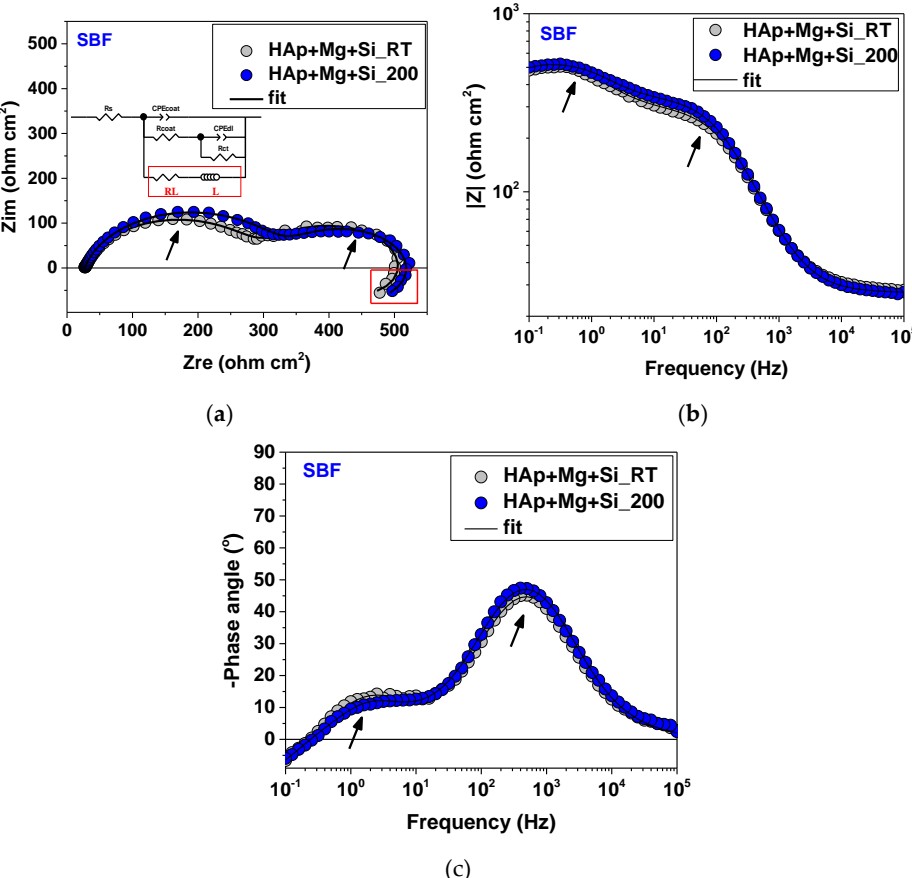

**Figure 6.** EIS curves of the investigated specimens immersed in SBF: (**a**) Nyquist plots, (**b**) |Z| plots, (**c**) Phase angle plots. ($R_s$ = the solution resistance, $CPE_{coat}$ = coating capacitance, $R_{coat}$ = resistance associated with the current flow, $CPE_{dl}$ = double layer capacitance, $R_{ct}$ = charge transfer resistance, L and RL were used to simulate the inductive behaviour).

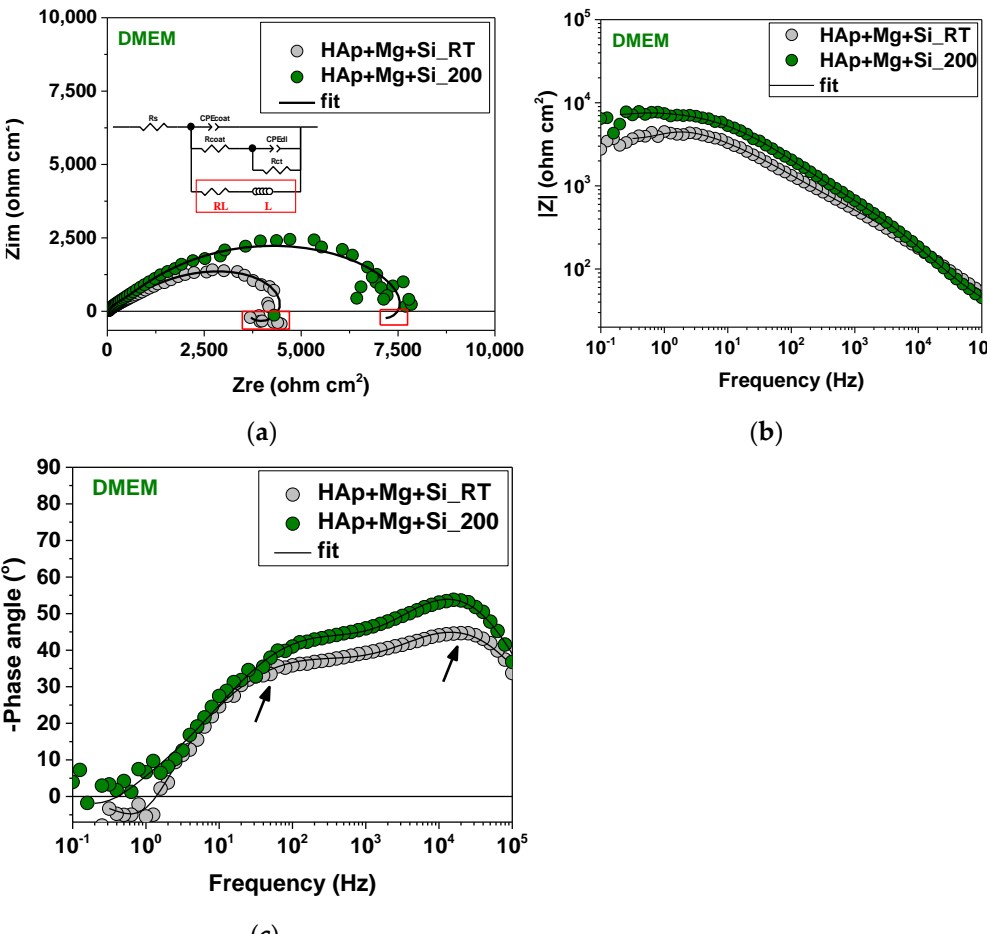

**Figure 7.** EIS curves of the investigated specimens immersed in DMEM: (**a**) Nyquist plots, (**b**) |Z| plots, (**c**) Phase angle plots. ($R_s$ = the solution resistance, $CPE_{coat}$ = coating capacitance, $R_{coat}$ = resistance associated with the current flow, $CPE_{dl}$ = double layer capacitance, $R_{ct}$ = charge transfer resistance, L and RL were used to simulate the inductive behaviour).

A comparative assessment of the HAp + Mg + Si corrosion protection ability as a function of the used electrolyte (SBF vs. DMEM) was further made based on the electrochemical parameters presented in Table 4. According to Orazem et al. [57–60], the quality of fitting can be evaluated by the $\chi^2$ error parameter, which in this case recorded values of $10^{-4}$–$10^{-3}$. As expected, HAp + Mg + Si showed superior electrochemical behaviour when immersed in a DMEM medium, and even higher dielectric properties were revealed when using an elevated temperature, i.e., 200 °C). Obviously, an increasing tendency of $R_{coat}$ can be observed, according to the coating deposition and immersion conditions, while the lowest value of $Q_{coat}$ was shown by HAp + Mg + Si_200 DMEM. At the electrolyte-substrate interface, similar values of double-layer capacitance ($Q_{dl}$ ~700 μF s($\alpha$ − 1) cm$^{-2}$) characterised the samples immersed in SBF, showing no improvement due to temperature in this case. However, the use of DMEM led to much lower values for H-Mg-Si RT ($Q_{dl}$ ~17 μF s($\alpha$ − 1) cm$^{-2}$) and HAp + Mg + Si_200 ($Q_{dl}$ ~7 μF s($\alpha$ − 1) cm$^{-2}$). The charge transfer reaction parameter was also influenced by the coating deposition and immersion conditions; the highest value obtained, and hence the best corrosion behaviour, was observed in the case of HAp + Mg + Si_200 DMEM ($R_{ct}$ = 42,274 Ω cm$^2$), proving a hindering effect in case of the mentioned coating, which limited the infiltration of the corrosive medium.

**Table 4.** EIS fitting results of the coatings obtained in SBF and DMEM solutions.

| Sample/Parameters | HAp + Mg + Si | | | |
|---|---|---|---|---|
| Temperature | RT | 200 | RT | 200 |
| Test Medium | SBF | | DMEM | |
| $R_s$ ($\Omega$ cm$^2$) | 28 | 27 | 26 | 25 |
| $Q_{coat}$ ($\mu$F s($\alpha - 1$) cm$^{-2}$) | 14.396 | 13.646 | 2.339 | 0.813 |
| $\alpha_{coat}$ | 0.84 | 0.84 | 0.70 | 0.79 |
| $R_{coat}$ ($\Omega$ cm$^2$) | 277 | 319 | 558 | 705 |
| $Q_{dl}$ ($\mu$F s($\alpha - 1$) cm$^{-2}$) | 757.480 | 749.400 | 17.533 | 7.459 |
| $\alpha_{dl}$ | 0.82 | 0.86 | 0.55 | 0.62 |
| $R_{ct}$ ($\Omega$ cm$^2$) | 235 | 193 | 5476 | 7624 |
| RL | 1219 | 1099 | 9664 | 42,274 |
| L | 3187 | 4083 | 1899 | 25,337 |
| $\chi2$ | $6 \times 10^{-4}$ | $7 \times 10^{-4}$ | $2 \times 10^{-3}$ | $4 \times 10^{-3}$ |

*3.5. In Vitro Degradation Tests*

Table 5 presents the samples mass loss expressed in mg, while Figure 8 presents the samples mass evolution (mass of the remained material along with the lost mass) expressed in percentages after each time interval (1, 3, 7 and 14 days), accompanied with macroscopic images of the samples.

**Table 5.** Mass loss of the coatings obtained in SBF and DMEM solutions.

| Coatings | Solutions | Day 1 | Day 3 | Day 7 | Day 14 |
|---|---|---|---|---|---|
| HAp + Mg + Si_RT | SBF | $-14.21 \pm 0.01$ | $-34.60 \pm 0.03$ | $-56.09 \pm 0.01$ | $-75.92 \pm 0.02$ |
| | DMEM | $-2.27 \pm 0.01$ | $-6.59 \pm 0.02$ | $-10.84 \pm 0.01$ | $-16.37 \pm 0.02$ |
| HAp + Mg + Si_200 | SBF | $-6.,72 \pm 0.01$ | $-48.34 \pm 0.,02$ | $-68.10 \pm 0.01$ | $-82.02 \pm 0.01$ |
| | DMEM | $-1.08 \pm 0.02$ | $-5.18 \pm 0.01$ | $-13.77 \pm 0.02$ | $-17.77 \pm 0.01$ |

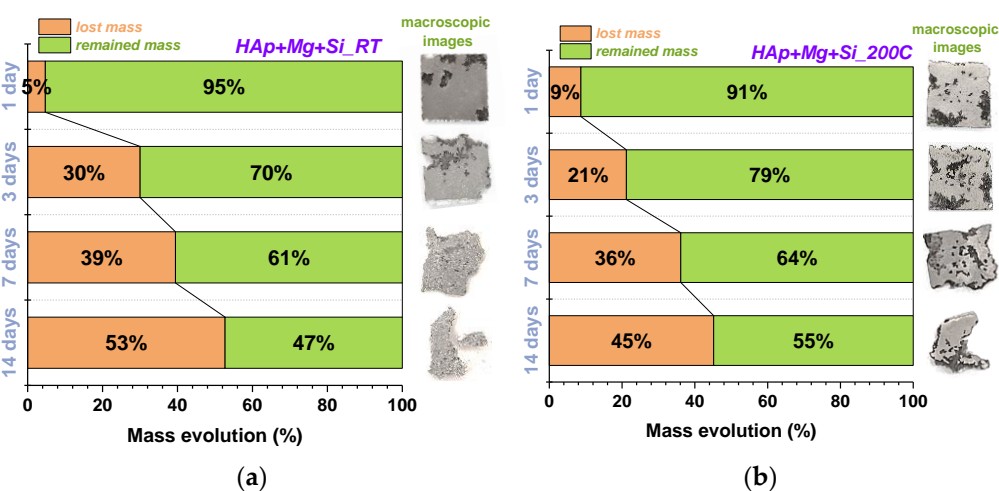

(**a**)　　　　　　　　　　　　　　　　　　　　(**b**)

**Figure 8.** *Cont.*

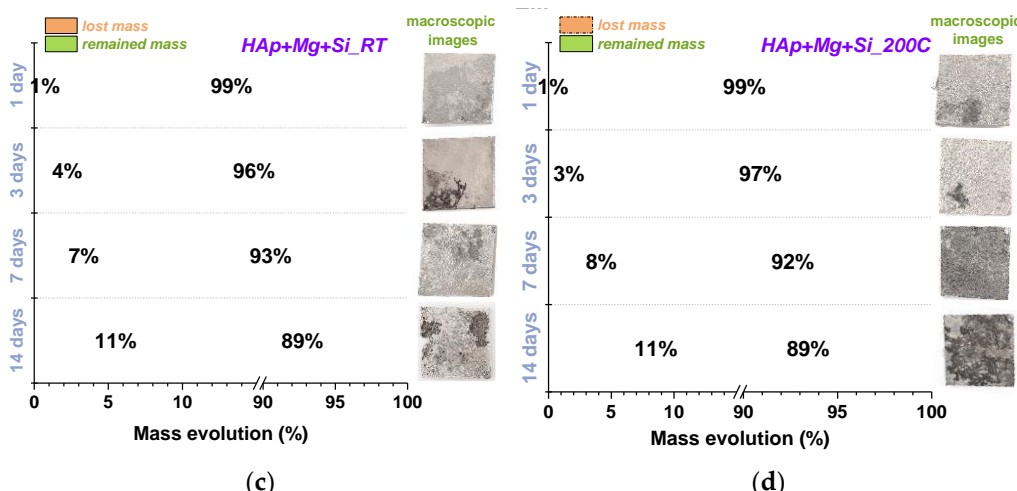

**Figure 8.** Evolution of mass during in vitro degradation tests: (**a**,**b**) Tests in SBF solution; (**c**,**d**) Tests in DMEM solutions, along with the optical images of the samples after the tests.

Figure 8 shows the results of the test in SBF and DMEM solutions presented in percentages for each immersion period to do a proper correlation of the results. The percentages were obtained by the ratio between the mass before and after tests.

Apparently, the highest degradation rate was recorded for the HAp + Mg + Si deposited at RT in the SBF medium, with a less aggressive degradation in DMEM.

After one day of immersion in SBF, the highest mass loss was noted for the coating deposited at 200 °C. After days 3, 7 and 14 of immersion in SBF, the degradation rate increased for the coatings deposited at RT, indicating a poor resistance in SBF at 37 °C. One may see in Figure 8 that a large surface of the samples deposited at 200 °C and immersed for 14 days in SBF is still there, compared with the surface of the coating prepared at RT. This finding shows that the increase in deposition temperature leads to a decrease in the degradation rate in SBF.

The immersion time in DMEM does not affect the degradation rate of investigated coatings, irrespective of the deposition temperature or immersion time. Both coatings exhibited similar degradation rates in DMEM solution (less than 2% differences in mass loss). Thus, it can be concluded that the investigated coatings are more resistant in DMEM solution compared with SBF.

Figure 9 presents the SEM images with the material's surface morphology after performing the immersion assays in both testing media, SBF and DMEM, respectively. According to the SEM images, it can be noted that in SBF, both coatings begin to present cracks after three days of immersion. The cracks were visible, irrespective of the immersion period. Starting the seventh day of immersion, on the material's surface, some small deposits start to appear, which after 14 days of immersion tend to agglomerate. This can be explained through the competition between biomineralisation (precipitation of some apatite) and biodegradation, a process known as bioactivity. This hypothesis corroborated with the results presented in Table 5 and Figure 8 highlights that, in this case, the degradation process is accelerated by the presence of Mg in the coating but also in the substrate. Moreover, as can be observed in Figure 9a, the presence of the cracks allows the SBF to reach the substrate, accelerating so the degradation of the experimental samples.

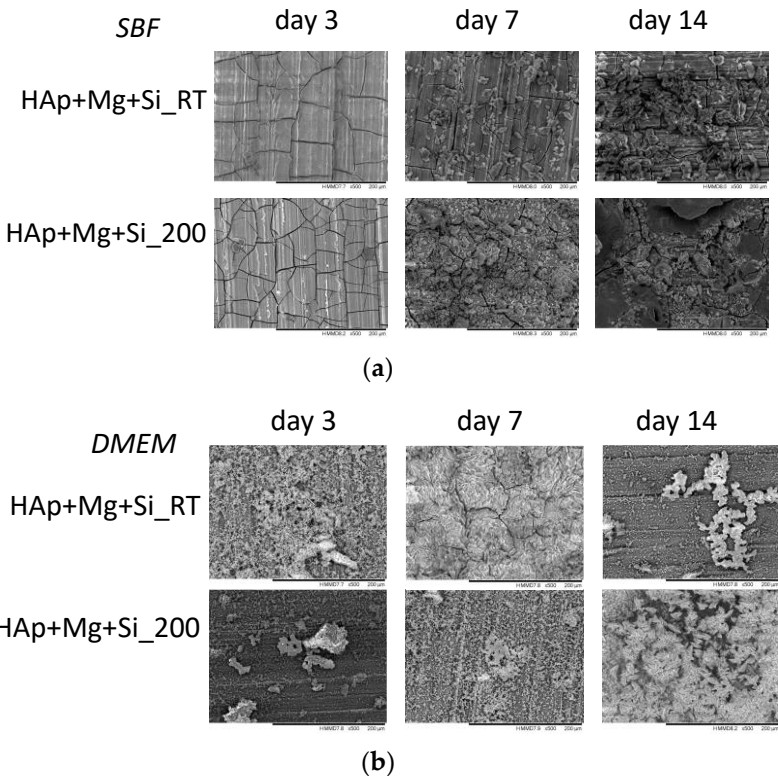

**Figure 9.** SEM images of degradation of surfaces after (**a**) Tests in SBF solution; (**b**) Tests in DMEM solutions.

In DMEM medium, which is less aggressive than SBF, the material surface does not generally present cracks and/or fissures, with one exception in the case of HAp + Mg + Si_RT, which after seven days of immersion, presented some surface alteration (some thin fissures can be observed along the material surface). These results are in agreement with the mass evolution ones presented in Figure 8, which have shown a lower degradation rate compared to the masses evolution in SBF.

*3.6. Surface Binding of Mussel-Derived Peptide to Coated AZ31B*

A biotin-based ELISA-like assay was performed to investigate the influence of the HAp-based coating as well as the deposition temperature on the binding property of the mussel-derived peptide (Figure 10). Mussle-derived peptides can be used to induce bio-integration, be further modified with cell recruiting or cell-binding segments.

Neither the coating composition nor the deposition temperature did affect the binding affinity of the peptide to the surface. Also, no significant difference between the control and the HAp + Mg + Si-coated samples has been detected.

Figure 10c,d displays 2D SEM images acquired for the mussel-derived peptide bound to the AZ31B substrates with HAp + Mg + Si_RT and HAp + Mg + Si_200 coatings deposited prior to peptide coating. In both cases, the SEM images show a similar coating of the material surface with the mussel-derived peptide, which is in line with the results obtained from the ELISA-like assay. Samples without previous HAp + Mg + Si coating (designated as uncoated) showed less peptide bound on the surface in both ELISA-like assay and SEM imaging (Figure 10b).

In conclusion, future bio-inspired surface-coating approaches do not limit the choice of prior coating compositions and temperatures.

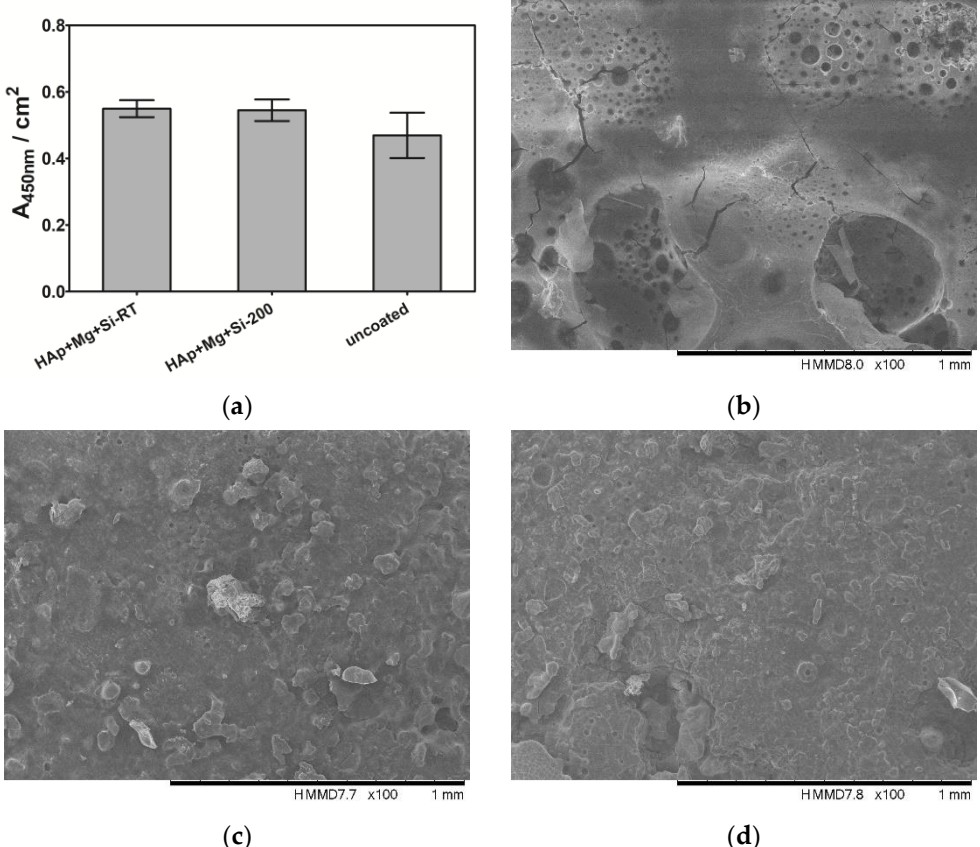

**Figure 10.** (**a**) Surface binding of the mussel-derived peptide to coated AZ31B. Samples were glued into wells of a 12-well plate prior to coating. Surface binding to uncoated AZ31B served as a control. Data are shown as mean $\pm$ standard error of the mean; $n$ = 2. (**b**–**d**) 2D SEM images of (**b**) no, (**c**) HAp + Mg + Si_RT and (**d**) HAp + Mg + Si_200 coatings deposited on AZ31B prior to coating with 1 $\mu$M mussel-derived peptide.

## 4. Conclusions

AZ31B alloys were coated with Mg and Si-doped HAp coatings at RT and 200 °C by RF magnetron sputtering method. The obtained results showed that incrementing the deposition temperature led to the formation of stoichiometric HAp structure irrespective of Mg and Si additions. As expected, the coatings prepared at 200 °C exhibited a denser structure with finer grains compared with ones deposited at RT, which reflected on the hardness and reduced modulus (~47% increase). Further on, a comparative assessment of corrosion resistance was made as a function of the immersion medium used, the coatings deposited at elevated temperature showing a superior corrosion resistance in both SBF and DMEM. However, the lowest $i_{corr}$ value (12 $\mu$A/cm$^2$) and the highest dielectric properties ($R_{coat}$ = 705 $\Omega$ cm$^2$, $R_{ct}$ = 7624 $\Omega$ cm$^2$) were obtained after immersion in DMEM of HAp + Mg + Si_200 coating, proving its protection ability against corrosion. Following the in vitro degradation tests, one can note that the investigated coatings presented a slower degradation rate in DMEM solution compared with SBF during 1, 3, 7 and 14 days of immersion at 37 °C. Moreover, an increment of the deposition temperature increases the resistance to degradation in SBF, while for the DMEM solution, the values are almost identical (less than 2% differences in mass loss). The mussel-derived peptide adhesion, as well as the SEM evaluation, indicated that the investigated coatings are suitable for further bio-inspired tests using mussel-derived peptides, regardless of the deposition temperature.

Importantly, based on the presented results, both coatings have improved the degradation of AZ31B alloy. More efficient protection was found for the coatings prepared at 200 °C, indicating that a higher deposition temperature is desirable when the degradation rate

should be slower. If a higher dissolution rate is required, the coatings prepared at RT should be used. For the biodegradable materials used in medical applications, proper control of the degradation rate is very important, and the selection of the coating is dependent on the desired effect, namely a slower or accelerated degradation rate. To conclude, both investigated coatings are proper to control the degradation rate of AZ31B biodegradable materials, but the use of one or another is dependent on the final application of the implant.

**Author Contributions:** The authors of the present paper had the following individual contributions: C.M.C., D.M.V., C.V.: Methodology, Validation, Writing—Review Editing; M.D.: XRD Investigation; A.V. Conceptualization; Methodology—Coatings preparation; Writing—Original Draft & Review Editing; Project administration; Funding acquisition; A.C.P.: Methodology; Data analysis; Writing—Review & Editing; M.D., D.M.V.: Writing—Original Draft; I.P.: Nanoindentation, FT-IR investigation; Data analysis; Writing—Original Draft; D.A.B.: Peptide synthesis and immobilisation, methodology and data analysis; A.G.B.-S.: Data analysis and Editing; G.S., G.A.: Alloy preparation, Writing—Review Editing. L.R.C. and I.M.M.: Investigation. All authors have read and agreed to the published version of the manuscript.

**Funding:** This work was funded by the joint research project "ISIDE—Innovative Strategies for bIoactive/antibacterial/advanceD prosthEses", a grant of (i) the Romanian National Authority for Scientific Research and Innovation, CCCDI—UEFISCDI, project number ERANET-M-ISIDE-1: 171/2020 (INOE) and ERANET-M-ISIDE-2: 172/2020 (UPB), within PNCDI III; (ii) the European Regional Development Fund, Calabria Region Grant J28I17000120005, M-ERA.Net 2—Call 2019, Italy, 2020–2023 and (iii) the Free State of Saxony mERA-NET ISIDE (ULEI) 100406843.

**Acknowledgments:** A.V.D. thanks to Tomsk Polytechnic University within the framework of the Tomsk Polytechnic University-Competitiveness Enhancement Program grant, as well to the Core Program within the National Research Development and Innovation Plan 2022–2027, carried out with the support of MCID, project no. PN 23 05 (id: PN11N-03-01-2023).

**Conflicts of Interest:** The authors declare no conflict of interest.

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
