# Peer review of "Effect of Deposition Temperature on the Structure, Mechanical, Electrochemical Evaluation, Degradation Rate and Peptides Adhesion of Mg and Si-Doped Hydroxyapatite Deposited on AZ31B Alloy"

_coatings, doi:10.3390/coatings13030591_

Round 1

Reviewer 1 Report

Authors presented the interesting research. The paper is suitable for publication after the little improvements. Superscripts and subsripts should be corrected in the lines 204, 205 and Table 3. Author used the same abbreviation SEM for both the scanning electron microscopy and standard error of the mean, it would be better in second case used another one. Sample designations in Fig. 1 b are different from presented in Table 1 and text.

Author Response

Dear Editor Puiu,

Thank you for your note and the reviewer comments on our manuscript. We would like to show our great gratitude to the editor and reviewer for the useful comments and constructive suggestions on our manuscript, which do help us significantly improve the quality of the current paper. All the review comments are appreciated. We do benefit a lot from the suggestions/ comments to improve the quality of our manuscript. We have revised our manuscript accordingly. The revision of the paper was highlighted by the blue coloured font. Detailed and point-to-point response to the reviewer comments is summarized below.

Here, we re-submit a new version of our manuscript which has been checked and modified after our careful referring to the reviewer comments. Meanwhile, efforts were also made to improve the English of the paper. We hope all of these changes will make this manuscript accepted by reviewers. Thank you for your kind consideration.

Best regards,

Alina Vladescu

REVIEWER 1

Comments and Suggestions for Authors

Authors presented the interesting research. The paper is suitable for publication after the little improvements. Superscripts and subsripts should be corrected in the lines 204, 205 and Table 3.

Thank you for pointing this out. Superscripts and subscripts were corrected through the manuscript.

Author used the same abbreviation SEM for both the scanning electron microscopy and standard error of the mean, it would be better in second case used another one.

The authors appreciate the observation. For clarification purposes, standard error of the mean was used in full name. In this manner the confusion with scanning electron microscopy will be avoided. 

Sample designations in Fig. 1 b are different from presented in Table 1 and text.

We are sorry about this error. In Fig. 1a, the designation was different indeed as compared with the ones used in Table 1 and through all the manuscript. Thus, the designations were corrected in whole manuscript to be similar.

Reviewer 2 Report

The reviewed manuscript entitled “Effect of deposition temperature and Si addition on the structure, mechanical, degradation and biocompatibility properties of Mg doped hydroxyapatite deposited on AZ31B alloy” investigates the enhancement ability of adding Si to the Mg doped hydroxyapatite deposited on AZ31B alloy at different temperatures. The article is made at a good scientific and technical level, and its practical significance is beyond doubt. In order to improve the readability and clarity of the manuscript, some major concerns need to be addressed before the paper is to be processed further:

1-  The manuscript is focusing mainly on the corrosion behavior, while the title is reflecting the effect on the structure, mechanical, degradation and biocompatibility properties. Please modify the title accordingly.

2-  The abstract requires some quantitative brief results. The abstract is a mini version of manuscript that proceeds. So, include introduction, methodology, results and concluding remarks in a precise but effective manner.

3- Avoid lumped references; a short comment should be included for each reference or two references in the same subject.

4-     The aim of the current study must be mentioned clearly at the end of introduction part specifying the objective and novelty of this work.

5- The motivation for the study and the research gap are not clear enough. Please demonstrate in the introduction of the paper, the novelty of this research in relation to other thematically similar research papers such as https://doi.org/10.1016/j.ceramint.2021.12.258., which published recently by the same authors.

6- Figure 1.a: Please provide the EDS spectrum for the different samples before extracting the shown elemental percentages. The raw spectrum is giving a full idea about the other elements and the contaminations as well.

7- Figure 1.b, GIXRD: Please identify all peaks in the XRD patterns, providing the unique powder diffraction file (PDF) or JCPDS Card Number for each element/compound/phase in the pattern. According to the EDS, the Mg is near 70% in some samples, which must be reflected in the X-ray diffraction pattern.

8- Figure 1.b, GIXRD: Why GIXRD was performed for the two Si substrate only ignoring the other two Mg substrate samples?

9- Discussion is lack of scientific explanation for the obtained results. Authors should attribute the results achieved to a clear scientific reason.

10- Please include some quantitative results from the study findings in the conclusion.

11- Remove the duplicated references in the references list (ex: Ref. 34 and Ref. 41).

12- The English language used in the paper is to be revised and improved before the subsequent manuscript submission. Please, read the text carefully before the next submission of the paper.

Author Response

Dear Editor Puiu,

Thank you for your note and the reviewer comments on our manuscript. We would like to show our great gratitude to the editor and reviewer for the useful comments and constructive suggestions on our manuscript, which do help us significantly improve the quality of the current paper. All the review comments are appreciated. We do benefit a lot from the suggestions/ comments to improve the quality of our manuscript. We have revised our manuscript accordingly. The revision of the paper was highlighted by the blue coloured font. Detailed and point-to-point response to the reviewer comments is summarized below.

Here, we re-submit a new version of our manuscript which has been checked and modified after our careful referring to the reviewer comments. Meanwhile, efforts were also made to improve the English of the paper. We hope all of these changes will make this manuscript accepted by reviewers. Thank you for your kind consideration.

Best regards,

Alina Vladescu

REVIEWER 2

Comments and Suggestions for Authors

The reviewed manuscript entitled “Effect of deposition temperature and Si addition on the structure, mechanical, degradation and biocompatibility properties of Mg doped hydroxyapatite deposited on AZ31B alloy” investigates the enhancement ability of adding Si to the Mg doped hydroxyapatite deposited on AZ31B alloy at different temperatures. The article is made at a good scientific and technical level, and its practical significance is beyond doubt. In order to improve the readability and clarity of the manuscript, some major concerns need to be addressed before the paper is to be processed further:

1-  The manuscript is focusing mainly on the corrosion behavior, while the title is reflecting the effect on the structure, mechanical, degradation and biocompatibility properties. Please modify the title accordingly.

Thank you for this suggestion. The title was modified accordingly.

2-  The abstract requires some quantitative brief results. The abstract is a mini version of manuscript that proceeds. So, include introduction, methodology, results and concluding remarks in a precise but effective manner.

Thank you for this suggestion. The abstract was modified accordingly.

3- Avoid lumped references; a short comment should be included for each reference or two references in the same subject.

Thank you for this suggestion. We have tried to do as you suggested. We hope all of these changes will be accepted by reviewer.

4-     The aim of the current study must be mentioned clearly at the end of introduction part specifying the objective and novelty of this work.

Details related to the aim of the study were added at the end of the introduction part.

5- The motivation for the study and the research gap are not clear enough. Please demonstrate in the introduction of the paper, the novelty of this research in relation to other thematically similar research papers such as https://doi.org/10.1016/j.ceramint.2021.12.258., which published recently by the same authors.

For clarification purposes, comparison with the mentioned study was made and details related to the objective of the paper were added in the revised version of the manuscript.

6- Figure 1.a: Please provide the EDS spectrum for the different samples before extracting the shown elemental percentages. The raw spectrum is giving a full idea about the other elements and the contaminations as well.

Thank you for this suggestion. We have added the EDS spectrum of each coating deposited on Si and Mg alloy – Figure 1 b.

7- Figure 1.b, GIXRD: Please identify all peaks in the XRD patterns, providing the unique powder diffraction file (PDF) or JCPDS Card Number for each element/compound/phase in the pattern. According to the EDS, the Mg is near 70% in some samples, which must be reflected in the X-ray diffraction pattern.

      For the detection of Mg content in the investigated coatings, the Si wafers substrates were used for EDS, showing ~7 at.% of Mg content. Thus, the influence of the Mg based substrate on the final chemical composition of the coatings was avoided. The mentioned low amount of Mg was not able to be identified when analysing the GIXRD patterns, the small peak located at 44.32° being attributed to HAp phase, according to JCPDS card no. 09-0432.

8- Figure 1.b, GIXRD: Why GIXRD was performed for the two Si substrate only ignoring the other two Mg substrate samples?

      As mentioned in the manuscript, the investigations were performed on the coatings deposited on Si wafers substrate for determining the Mg elemental composition and on Mg alloy substrate for determining the Si content (to avoid the influence of the Mg or Si element from each substrate). Regarding the GIXRD results, it was not reasonable to investigate the coated Mg substrate samples, since there was no additional information to be added.

9- Discussion is lack of scientific explanation for the obtained results. Authors should attribute the results achieved to a clear scientific reason.

Thank you for this suggestion. We have improved the explanation on the manuscript in all section. We hope all of these changes will be accepted by reviewer.

10- Please include some quantitative results from the study findings in the conclusion.

Thank you for this suggestion. The abstract was modified accordingly.

11- Remove the duplicated references in the references list (ex: Ref. 34 and Ref. 41).

We are sorry about these errors. The references were added automatically, and we do not know why was not updated. We have corrected this error.  

12- The English language used in the paper is to be revised and improved before the subsequent manuscript submission. Please, read the text carefully before the next submission of the paper.

The authors appreciate the observation. The manuscript was verified and grammatical errors were corrected by a language editing services.

Reviewer 3 Report

Dear authors

Manuscript is well return  and results ad discussion part was interesting. But some more quiries needs to be addressed before accpetance.

Abstract and conclusion part should fallow the same path and it needs to be reframed.

Novality of the work should be clearly explained in the abstract and introduction part.

Author explained mechanical characterisation in abstract ( Which mechanical property, needs to be specified in the abstract)

What is the % of improvement in the property , not explained in the manuscript.

Lot of typo errors were observed, like 1.8 gm/cm3 (Subscript and superscript)

Table 1, repeated two times (line# 132 and 167)

Table 1- line # 132, what is the standard wt.% of material, needs to be shown in separate row.

2.1, 2.2 etc., sub headings seems to be general name, it should be reframed.

Paper should be throughly checked for grammatical mistakes.

Author Response

Dear Editor Puiu,

Thank you for your note and the reviewer comments on our manuscript. We would like to show our great gratitude to the editor and reviewer for the useful comments and constructive suggestions on our manuscript, which do help us significantly improve the quality of the current paper. All the review comments are appreciated. We do benefit a lot from the suggestions/ comments to improve the quality of our manuscript. We have revised our manuscript accordingly. The revision of the paper was highlighted by the blue coloured font. Detailed and point-to-point response to the reviewer comments is summarized below.

Here, we re-submit a new version of our manuscript which has been checked and modified after our careful referring to the reviewer comments. Meanwhile, efforts were also made to improve the English of the paper. We hope all of these changes will make this manuscript accepted by reviewers. Thank you for your kind consideration.

Best regards,

Alina Vladescu

Comments and Suggestions for Authors

Dear authors

Manuscript is well return  and results ad discussion part was interesting. But some more quiries needs to be addressed before accpetance.

Abstract and conclusion part should fallow the same path and it needs to be reframed.

Thank you for this suggestion. The abstract and the conclusions were reframed.

Novality of the work should be clearly explained in the abstract and introduction part.

The novelty of the paper was highlighted in the revised version of the manuscript.

Author explained mechanical characterisation in abstract ( Which mechanical property, needs to be specified in the abstract)

The abstract was modified and the type of mechanical characterization was specified.

What is the % of improvement in the property , not explained in the manuscript.

The comment in the abstract related to the % improvement in hardness and degradation rate with Si addition into Mg doped hydroxyapatite coatings was based on the information found in literature. The sentence was deleted since this result does not represent an outcome of the current paper. The abstract was modified according to the its content.

Lot of typo errors were observed, like 1.8 gm/cm3 (Subscript and superscript)

Thank you for pointing this out. Superscripts and subscripts were corrected through the manuscript.

Table 1, repeated two times (line# 132 and 167)

The order of the tables in the manuscript was corrected.

Table 1- line # 132, what is the standard wt.% of material, needs to be shown in separate row.

The chemical composition presented in Table 1 was given in wt.%. The missing information was added.

2.1, 2.2 etc., sub headings seems to be general name, it should be reframed.

As stated in Instructions for Authors, the headings of the main sections should give a clear insight related to the description of the experimental materials, methods or results. Considering this information, the authors formulated the subheadings according to the content.  

Paper should be throughly checked for grammatical mistakes.

The authors appreciate the observation. The manuscript was verified and grammatical errors were corrected by a language editing services.

Round 2

Reviewer 2 Report

The revision is satisfactory and the authors have provided amendments to all the suggested queries. Therefore, I recommend this work for publication in Coatings.